# Importance of Class Selectivity in Early Epochs of Training

## Abstract

Deep networks trained for classification exhibit class-selective neurons in intermediate layers. Intriguingly, recent studies have shown that class-selective neurons are not strictly necessary for network function. But if class-selective neurons are not necessary, why do they exist? We attempt to answer this question in a series of experiments on ResNet-50 trained on ImageNet. We begin by showing that class-selective neurons emerge in the first few epochs of training before receding rapidly. Single-neuron ablation experiments show that class-selective neurons are important for network function during this early phase of training. The network is close to a linear regime during this early training phase, which may explain the emergence of these class-selective neurons in intermediate layers. Finally, by regularizing against class selectivity at different points in training, we show that the emergence of these class-selective neurons during the first few epochs of training is essential to the successful training of the network. Altogether, our results indicate that class-selective neurons in intermediate layers are vestigial remains of early epochs of training, during which they appear as quasi-linear shortcut solutions to the classification task which are essential to the successful training of the network.

## 1 Introduction

A significant body of research has attempted to understand the role of single neuron class-selectivity in the function of artificial (Zhou et al., 2015; Radford et al., 2017; Bau et al., 2017; Morcos et al., 2018; Olah et al., 2018; Rafegas et al., 2019; Dalvi et al., 2019; Meyes et al., 2019; Dhamdhere et al., 2019; Leavitt & Morcos, 2020a; Kanda et al., 2020; Leavitt & Morcos, 2020b), and biological (Sherrington, 1906; Adrian, 1926; Granit, 1955; Hubel & Wiesel, 1959; Barlow, 1972) neural networks. Neurons responding selectively to specific classes are typically found throughout networks trained for image classification, even in early and intermediate layers. Interestingly, these class-selective neurons can be ablated (i.e. their activation set to 0; Morcos et al. 2018) or class selectivity substantially reduced via regularization Leavitt & Morcos (2020a) with little consequence to overall network accuracy—sometimes even improving it. These findings demonstrate that class selectivity is not necessary for network function, but it remains unknown *why* class selectivity is learned if it is largely not necessary for network function.

One notable limitation of many previous studies examining selectivity is that they have largely overlooked the temporal dimension of neural network training; single unit ablations are performed only at the end of training (Morcos et al., 2018; Amjad et al., 2018; Zhou et al., 2018; Meyes et al., 2019; Kanda et al., 2020), and selectivity regularization is mostly constant throughout training (Leavitt & Morcos, 2020a;b). However, there are numerous studies demonstrating substantial differences in training dynamics during the early vs. later phases of neural network training (Sagun et al., 2018; Gur-Ari et al., 2018; Golatkar et al., 2019; Frankle et al., 2020b; Jastrzebski et al., 2020). Motivated by these studies, we asked a series of questions about the dynamics of class selectivity during training in an attempt to elucidate *why* neural networks learn class selectivity: When in training do class-selective neurons emerge? *Where* in networks do class-selective neurons first emerge? Is class selectivity uniformly (ir)relevant for the entirety of training, or are there "critical periods" during which class selectivity impacts later network function? We addressed these questions in experiments conducted in ResNet-50 trained on ImageNet, which led to the following results:

- The emergence of class-selective neurons in early and intermediate layers follow a non-trivial and surprising path: after a prominent rise during the first few epochs of training, class selectivity subsides quickly during the next few epochs, before returning to a baseline level specific to each layer.

- During this early training phase where average class selectivity is high in early and intermediate layers, class-selective neurons in these layers are much more important for network function compared to later in training, as assessed with single-unit ablation.

- During this early, high-selectivity phase of training, the representations of early and latter layers are much more similar than during later in training, implying that selectivity in early layers could be leveraged to solve the classification problem by transmission to the latter layers via skip connections.

- In a causal experiment where we prevent class selectivity from rising sharply in early and intermediate layers during the first epochs of training, we show the network training accuracy suffers from the suppression of this phenomenon. This indicates that the rapid emergence of class-selective neurons in early and intermediate layers during the first phase of training plays an essential role in the successful training of the network.

In conclusion, class-selective neurons in early and intermediate layers of deep networks seem to be a vestige of their emergence during the first few epochs of training, during which they play an essential role to the successful training of the network.

## 2 RELATED WORK

### 2.1 THE ROLE OF SELECTIVITY IN DEEP NETWORKS

Numerous studies have examined the causal role of class selectivity for network performance, nearly all of which have relied on single unit ablation as their method of choice. Morcos et al. 2018 examined a number of different CNN architectures trained to perform image classification and found that class selectivity was uncorrelated (or negatively correlated) with test-set generalization. This finding was replicated by Kanda et al. 2020 and Amjad et al. 2018, though the latter study also observed that the effects can vary when ablating groups of neurons, in which case, selectivity can be beneficial. Furthermore, Zhou et al. 2018 found that ablating class-selective units can impair accuracy for specific classes, but a corresponding increase in accuracy for other classes can leave overall accuracy unaffected.

Studies using NLP models have also shown varied results. Donnelly & Roegiest 2019 ablated the "sentiment neuron" reported by Radford et al. 2017 and found mixed effects on performance, while Dalvi et al. 2019 found networks were more negatively impacted when class-selective units were ablated.

Of particular importance is the work of Leavitt & Morcos 2020a, which introduced a regularizer for fine-grained control over the amount of selectivity learned by a network. This regularizer makes it possible to test whether the presence of selectivity is beneficial, and whether networks need to learn selectivity, two questions that ablation methods cannot test. By regularizing to discourage or promote the learning of selectivity, they found that selectivity is neither strictly necessary nor sufficient for network performance. In follow-up work they also showed that promoting class selectivity with their regularizer confers robustness to adversarial attacks, while reducing selectivity confers robustness to naturalistic perturbations (Leavitt & Morcos, 2020b). However, they did not scale their experiments beyond ResNet18 (He et al., 2016) and Tiny-ImageNet (Fei-Fei et al., 2015). Additionally, with the exception of one set of controls in which the regularizer was linearly warmed up during the first five epochs of training (Leavitt & Morcos, 2020a), they did not examine the dynamics of class selectivity's importance over the training process. Most importantly, they did not attempt to address *why* selectivity is learned.

### 2.2 THE EARLY PHASE OF NEURAL NETWORK TRAINING

A breadth of approaches have been used to characterize the differences between the early and later phases of neural network training and highlight the impact of early-phase interventions on late-phase

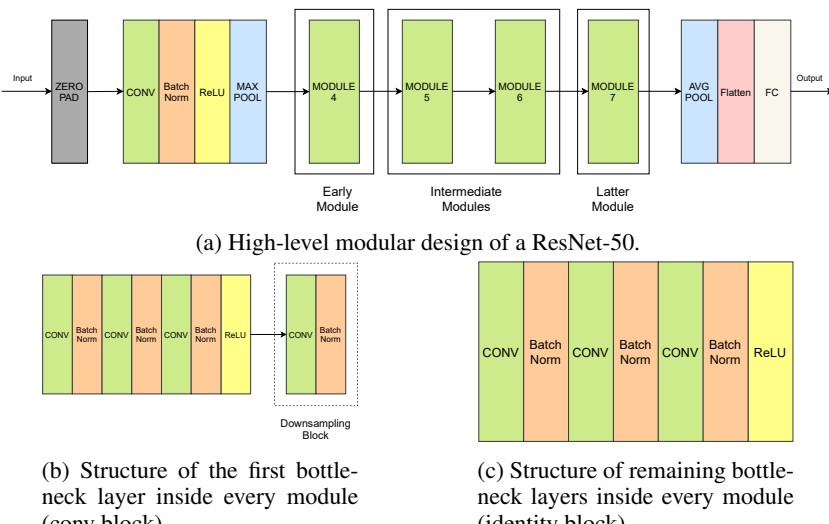

(a) High-level modular design of a ResNet-50.

(b) Structure of the first bottle-neck layer inside every module (conv block).

(c) Structure of remaining bottle-neck layers inside every module (identity block).

Figure 1: **Structure of ResNet-50. (a)** We define the early and intermediate layers as the layers present in the early and intermediate modules. **(b, c)** Each of these modules are made of multiple bottleneck layers which are also commonly known as conv block and identity block in ResNets.

behavior. The application of iterative magnitude pruning in the Lottery Ticket approach (Frankle & Carbin, 2018) requires rewinding to a sufficiently early point in training—within the first few thousand iterations—for any benefit to be realized (Frankle et al., 2020a;b). The local loss landscape also changes rapidly early in training (Sagun et al., 2018); the subspace in which gradient descent occurs quickly shrinks into a very restricted subspace (Gur-Ari et al., 2018). Achille et al. 2018 characterized a critical period of training by perturbing the training process with corrupted data labels early in training and found that the network's final performance was irreparably impaired. Similarly, Golatkar et al. 2019 found that removing some forms of regularization after the early phase of training, or imposing them after the early phase, had little effect on network performance. These results emphasize the outsize importance of interventions applied during the early phase of neural network training, and the relevance of the early phase of training for understanding why networks learn class selectivity.

## 3 APPROACH

### 3.1 MODEL ARCHITECTURE

We use ResNet-50s (He et al., 2016) trained on ImageNet for all our experiments. ResNet-50s are formed of a convolutional stem followed by four successive modules. We define the early and intermediate layers as the layers present inside the early and intermediate modules shown in Fig 1a. Each module is made up of bottleneck layers whose structure is shown in Fig 1b and Fig 1c. Bottleneck layers are the layers where the residual connections happen. Details on model training can be found in Appendix A.1.

### 3.2 CLASS SELECTIVITY INDEX

We refer to the individual channels of the bottleneck layers as "units" or "neurons" for the purpose of our experiments. We compute a class selectivity index for each unit of each bottleneck layer of every module. This index is a measure of how much the unit favors a specific ImageNet class. It is defined as (Leavitt & Morcos, 2020a):

$$SI = \frac{\mu_{max} - \mu_{-max}}{\mu_{max} + \mu_{-max} + \epsilon} \tag{1}$$

where $\mu_{max}$ is the largest class-conditional mean activation over all possible classes, averaged within samples of a class, and averaged over spatial locations for that channel, $\mu_{-max}$ is the mean response to the remaining (i.e. non-$\mu_{max}$) classes, and $\epsilon$ is a small value to prevent division by zero (here $10^{-6}$) in the case of a dead unit. All activations are assumed to be positive because they are extracted after a ReLU layer. The selectivity index can range from 0 to 1. A unit with identical average activity for all classes would have a selectivity of 0, and a unit that only responded to a single class would have a selectivity of 1. We compute such class selectivity indices across all epochs of training and then we perform various experiments to see how class selectivity affects learning at different stages of training.

### 3.3 QUANTIFYING THE RELEVANCE OF CLASS-SELECTIVE NEURONS

We used the following methods to determine whether class-selective neurons are relevant to network function.

**Single neuron ablation:**    This method consists in setting the channel outputs of the bottleneck layers to 0. In our experiments, we perform progressive ablation of the channels in all bottleneck layer of a given module, ordered by class selectivity index and starting with the most class-selective channels, or by random ordering of channels (random control condition). On a fully-trained ResNet-50, we find that ablation of class-selective neurons is less damaging to network accuracy than ablation of random neurons (Appendix A.2), consistent with previous reports.

**Class-selectivity regularizer:**    We used Leavitt & Morcos (2020a)'s selectivity regularizer to either promote or suppress the emergence of class-selective neurons in different modules and at different epochs of training. The regularization term is as follows:

$$\mu_{SI} = \frac{1}{M} \sum_m^M \frac{1}{B} \sum_b^B SI_b \tag{2}$$

where M is the total number of modules, m is the current module, B is the total number of bottleneck layer in current module M and b is the current bottleneck layer. $SI_b$ represents the selectivity indices of the channels in the current bottleneck layer b.

This term can be added to the loss function as follows:

$$loss = -\sum_c^C y_c \log(\hat{y_c}) - \alpha\mu_{SI} \tag{3}$$

The left hand term of the loss function is the cross-entropy loss where C is the total number of classes, c is class index, $y_c$ is true class label and $\hat{y_c}$ is the predicted class probability. The right hand side of the loss function is the regularization term ($-\alpha\mu_{SI}$). The alpha can be used to control the strength of the regularizer. A negative value of $\alpha$ will regularize against selectivity and a positive value of $\alpha$ will regularize in favor of selectivity.

## 4 RESULTS

### 4.1 CLASS-SELECTIVE NEURONS EMERGE DURING EARLY EPOCHS OF TRAINING

We began by examining the dynamics of class selectivity in each module over the course of training. It's possible that selectivity follows a similar pattern as accuracy: increasing quickly over the first few epochs, then gradually saturating over the remainder of training. We consider this to be the most likely scenario, as it is follows the dynamics of early and rapid change followed by stabilization observed in many other quantities during training. Indeed, class selectivity sharply rises during the first epoch of training (Fig 2; analysis of evolution of selectivity throughout training for individual bottleneck layers can be found in Appendix A.3), but after that, the effect varies depending on layer depth: earlier layers (Modules 4-6) show rapid decreases in selectivity over 3-5 epochs, followed by slow approach to an asymptote over the remainder of training. In contrast, latter layers (Module 7) do not exhibit any substantial decrease in selectivity after the initial jump, and selectivity remains

high in latter layers throughout training. These dynamic, depth-dependent changes in selectivity during the first few epochs of training indicate that the role of selectivity could also vary during training, and in a layer-dependent manner.

### 4.2 CLASS-SELECTIVE NEURONS SUPPORT THE DECISION OF THE NETWORK MORE DURING EARLY EPOCHS OF TRAINING THAN LATTER EPOCHS

In order to determine whether the role of class selectivity in network behavior varies over the course of training, we performed class-selective and random ablations (Section 3.3) at different points during training and measured the effect on accuracy. We normalize accuracy between 0 and 100 so that the curves for each epoch can be plotted together and compared. The right-hand side of Fig 3 shows that the effect on accuracy is more prominent in the earlier epochs of class-selective ablations. Compared to this, random ablation curves do not show any significant difference across epochs.

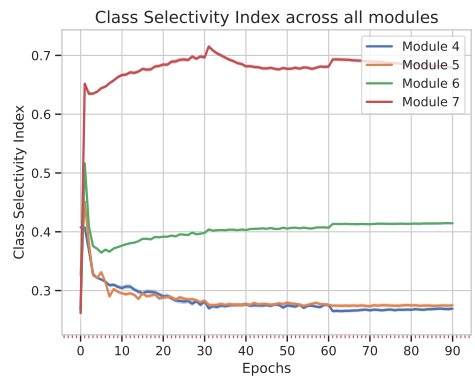

Figure 2: **Evolution of class selectivity throughout training in all modules of a ResNet-50.** Each line represents the evolution of average class selectivity of neurons of a given module throughout training for a ResNet-50 trained on ImageNet. A sharp rise in selectivity can be seen in all modules during the first few epochs of training, which recedes rapidly for the intermediate modules (modules 4,5,6) but not for the last module (module 7).

To quantify this phenomenon further, we calculated the area under the ablation curves for each epoch, obtaining one value per epoch summarizing the sensitivity of the network to ablation, and plotted these values across epochs. If class-selective ablations were more damaging in the earlier epochs, then this would correspond to less area under the curve (AUC) in the earlier epochs compared to the latter epochs i.e., the curve should show a positive slope across epochs. As illustrated in Fig 3, the class-selective curves for module 4 and module 6 indeed show a positive slope which is steeper especially at early epochs compared to the random ablation curve which is relatively flat. These results indicate that class-selective neurons are more important in the earlier epochs of training compared to the latter epochs as they are more damaging to the normalized accuracy in the earlier epochs compared to latter epochs. Interestingly, module 5 doesn't exhibit this trend as clearly.

However, the AUC curves in Fig 3 also show that random ablations remain more damaging than class-selective ablations at all epochs, as they exhibit overall less area under the curve. The important distinction is that class-selective ablations are more damaging in the earlier epochs *relative to the class-selective ablations in latter epochs* while random ablations are roughly equally damaging throughout training. This phenomenon reverses in module 7 (Fig 4), where class-selective ablations are more damaging throughout training, which is expected from the role of these layers to provide class-selective information for the network decision.

### 4.3 THE NETWORK IS LINEAR AT INITIALIZATION AND IN EARLY EPOCHS OF TRAINING, WHICH MIGHT EXPLAIN THE EMERGENCE OF SHORTCUT DECISION STRATEGIES

Why are neurons learning class-selective features in early and intermediate modules during the early epochs of training? Here, we postulate that the network is close to being linear at random weight initialization, such that the neurons of all layers jointly learn to be class-selective, with neurons from latter layers relying on class-selective features emerging in early and intermediate layers, effectively producing crude shortcut decision strategies at these early epochs of training.

To test out this hypothesis, we calculated the Centered Kernel Alignment (CKA) (Kornblith et al., 2019; Nguyen et al., 2020) score between all modules. CKA can be used to find representational similarity between different components of a network. The CKA score between early module and latter module and between early module and the fully connected layer (FC) is indeed high during

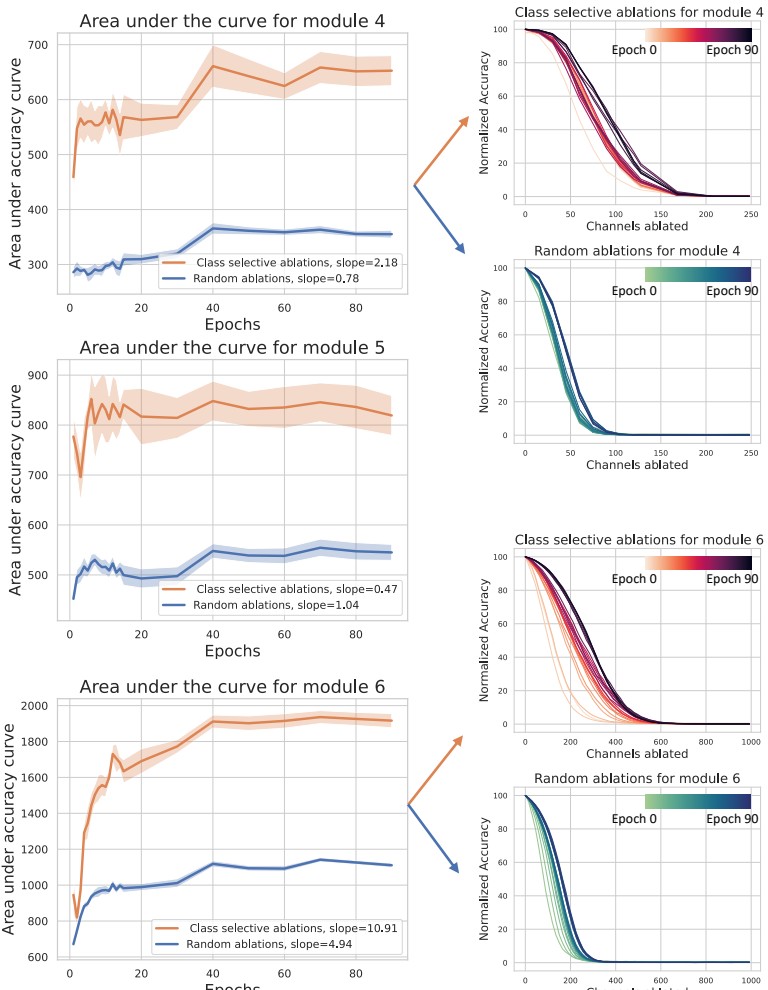

Figure 3: **Effect of class-selective ablations and random ablations on normalized network accuracy throughout training for early and intermediate modules.** The right-hand side of the figure depicts the effect of class-selective neurons ablations on network accuracy throughout epochs. The effect on accuracy is more damaging during the earlier epochs of class-selective ablations compared to latter epochs. The random ablation curves do not show this effect (curves are close to each other). The left-hand side of the figure depicts the area under the ablation curves (AUC), as calculated by the sum of accuracies for each epoch-specific curve shown on the right-hand side. The positive slopes of the AUC curves for the class-selective ablations, especially visible during the early epochs of training, indicate that class-selective ablations are more important for network decision during these early epochs compared to latter epochs. Error shades indicate the 95% CI of the mean, calculated over 10 networks trained from different random initial conditions.

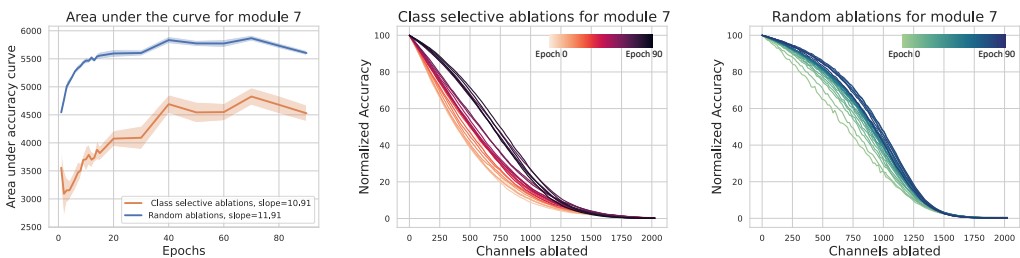

Figure 4: **Effect of class-selective ablations on the latter module.** The effect of class-selective ablations is different in module 7 than in the intermediate modules: class-selective neurons remain important throughout training, not just in the early epochs. Error shades indicate the 95% CI of the mean, calculated over 10 networks trained from different random initial conditions.

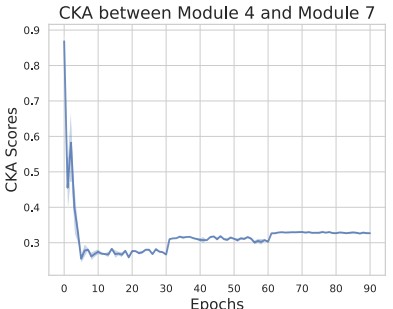 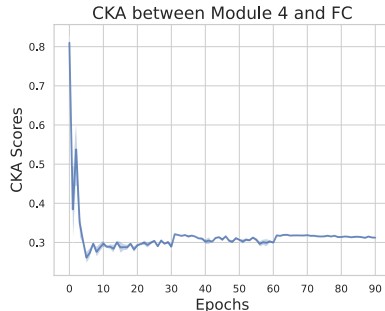

Figure 5: **Representational similarity between modules throughout training.** We measure the representational similarity between the early module 4 and latter layers (module 7 layers, fully connected layer (FC)) using CKA. We find that the similarity between these layers is high at the early epochs of training, indicating that the network is close to linear at this stage. This linearity might be key to explain the emergence of class-selective neurons in all layers during the early phase of training (see text). Error shades indicate the 95% CI of the mean, calculated over 6 networks trained from different random initial conditions.

the first few epochs, and then recedes to a low level for the rest of the training (Fig 5). The CKA score across all other pairs of modules shows a similar trend (Appendix A.4).

From these results, the following picture emerges: at the early stages of training where the network is close to being linear, all layers jointly learn class-selective features, with latter layers piggy-backing on class-selective features learned by early layers.

### 4.4 THE EMERGENCE OF CLASS-SELECTIVE NEURONS EARLY IN TRAINING IS ESSENTIAL TO SUCCESSFUL TRAINING: SUPPRESSING IT IMPAIRS LEARNING

Is the emergence of class-selective neurons early in training crucial to learning? To test whether the spike observed in selectivity during the early epochs (Fig 2) is important to learning, we conducted a series of regularization experiments. The question which we want to answer is - Will the model be able to learn if we regularize against selectivity and suppress that spike in selectivity in the early epochs?

We perform experiments to test two scenarios: regularizing against class-selectivity from epoch 0 onward VS regularizing from epoch 5 onward. Epoch 0 is the randomly initialized model and epoch 5 is approximately after the spike in selectivity (Fig 2). We hypothesize that if selectivity is important to learning in those early epochs, then regularizing from epoch 0 onward should lead to a greater decrease in the model accuracy later on in training as compared to regularizing from epoch 5 onward. As we already know that selectivity is important to learning in the latter module and as we are interested in understanding the emergence of class-selective neurons in early and intermediate modules, we only regularize the early and intermediate modules against selectivity.

We used a regularizing strength of $\alpha = -20$ and found that the regularizer prevents the peak of emergence of class-selective neurons in early epochs of training and that it impairs training and validation accuracy throughout training if regularized from epoch 0. However, in the second scenario where the regularizer was turned on from epoch 5 onward, the model performed almost as well as the original unregularized model (Fig 6). These results indicates that the emergence of class-selective neurons in intermediate modules in early epochs of training is an phenomenon that plays a crucial role in the correct training of the network. Other sets of experiments with different values of $\alpha$ can be found in Appendix A.5.1. It is worth noting that Leavitt & Morcos (2020a) found that regularizing beyond the range of $\alpha = [-1, -5]$—depending on the model and dataset—had significant negative effects on network performance, while we were able to increase the magnitude of $\alpha$ all the way to $-20$ with minimal effect on network performance, so long as we did not regularize until epoch 5.

The model never fully recovers the performance if the regularization is kept on, but if the regularization is turned off after first few epochs of training then the network is slowly able recover from the perturbation after 20+ epochs of training (Appendix A.5.3).

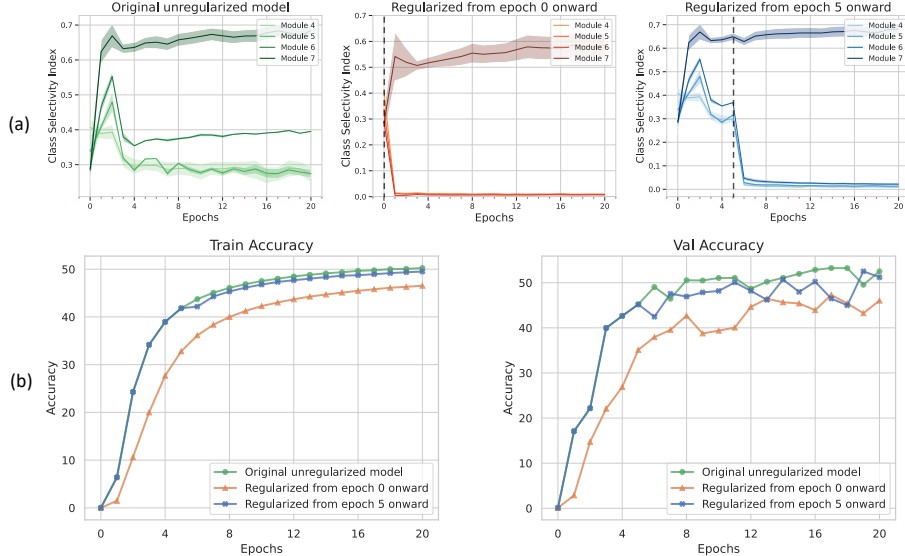

Figure 6: **Regularization against selectivity.** **(a)** Class selectivity indices throughout training for three models, respectively unregularized, regularized from epoch 0 onward, and regularized from epoch 5 onward. The effect of regularization on class-selectivity is clearly visible in the second and third plots. Note that only early and intermediate modules are regularized. Error shades indicate the 95% CI of the mean, calculated over all the bottleneck layers for each module. **(b)** Train and Validation accuracies for each model throughout training. The model regularized against selectivity from epoch 5 onward performs as well as the original unregularized model. The model regularized against selectivity from epoch 0 onward performs significantly worse than the other two models. This result shows the importance of class selectivity in the early epochs of training.

### 4.5    CLASS SELECTIVITY IS MOST IMPORTANT DURING THE FIRST EPOCH OF TRAINING

In the previous set of experiments, we regularized against class-selectivity from epoch 5 onward which was roughly after the spike in selectivity. But the next question which arises is what happens if we regularize in-between epochs 0 and 5? Fig 7 illustrates that learning is only impaired if the model is regularized from epoch 0 onward. Models regularized from epoch 1-5 onward performed similarly, suggesting that the most important phase of class selectivity is during the very first epoch.

### 4.6    REGULARIZING IN FAVOR OF CLASS SELECTIVITY DOES NOT IMPROVE LEARNING

As class-selective neurons are crucial to learning in the early epochs, could increasing the selectivity in those early epochs further improve learning? To answer this question, we regularized *in favor* of class-selectivity (i.e. with positive values of $\alpha$) in those early epochs. We found that increasing the selectivity in those early epochs did not improve performance further (Fig 13, Fig 14). Therefore, both an increase and a decrease in selectivity is harmful to learning, suggesting that the network roughly learns the "right" amount of selectivity for optimal learning. More details on the regularization for selectivity experiments can be found in Appendix A.5.2.

### 4.7    EFFECT OF CLASS SELECTIVITY REGULARIZATION ON THE BALANCE OF CLASS REPRESENTATION

We next explored how the regularization in favor and against class selectivity affects the balance of class representation in the predictions of the network. We found that, both increasing and decreasing class selectivity in early and intermediate modules resulted in a poorer diversity of class representation in the predictions of the network (Appendix A.5.4). This effect lasted for the few early first epochs of training, after which it was no longer visible. We conclude that tampering with class selectivity in early and intermediate layers perturbs the correct learning of the network by over-

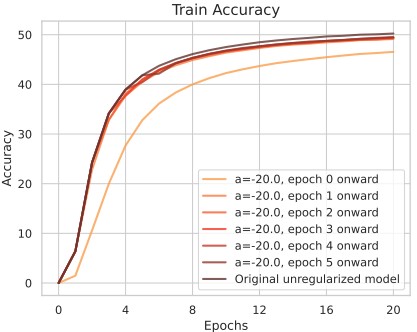 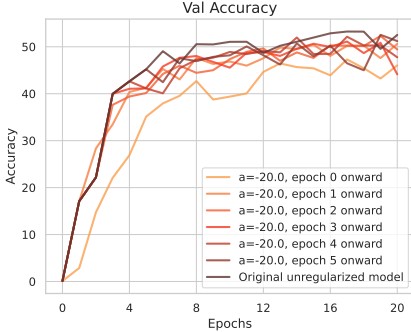

Figure 7: **Effect of regularization against selectivity when starting from different epochs.** Only the model regularized from epoch 0 onward performs poorly. Models which were regularized starting from epoch 1-5, all performed similarly. This suggests that the critical phase during which class-selectivity is important is restricted to the very first epoch of training.

representing certain classes at the prediction stage during a period which is apparently critical for the correct training of the network.

## 5 DISCUSSION

Previous experiments have shown that class selectivity in individual units may not be necessary for high accuracy converged solutions (Morcos et al., 2018; Amjad et al., 2018; Donnelly & Roegiest, 2019; Leavitt & Morcos, 2020a; Kanda et al., 2020; Leavitt & Morcos, 2020b). We sought to determine *why* class selectivity is learned if it is not necessary. In a series of experiments examining the training dynamics of ResNet50 trained on ImageNet, we found that class selectivity in early and intermediate layers is actually necessary during a critical period early in training in order for the network to learn high accuracy converged solutions. Specifically, we observed that class selectivity rapidly emerges within the first epoch of training, before receding over the next 2-3 epochs in early and intermediate layers. Ablating class-selective neurons during this critical period has a much greater impact on accuracy than ablating class-selective neurons at the end of training. Furthermore, we used Leavitt & Morcos (2020a)'s selectivity regularizer and found that strongly regularizing against class selectivity—to a degree that Leavitt & Morcos (2020a) found had catastrophic effects on network performance—had a negligible effect on network performance *if regularization doesn't occur in the initial epochs*. We also found that the representations of early and latter layers are much more similar during the early critical period of training compared to later in training, indicating that selectivity in early layers may be leveraged by latter layers in order to solve the classification problem early in training. Taken together, our results show that class-selective neurons in early and intermediate layers of deep networks are a vestige of their emergence during a critical period early in training, during which they are necessary for successful training of the network.

One limitation of the present work is that it is limited to ResNet-50 trained on ImageNet. While Leavitt & Morcos (2020a) found that their results were consistent across a breadth of CNN architectures and image classification datasets, it's possible that our findings may not generalize to networks trained on NLP tasks, in which single neuron selectivity is also a topic of interest.

Our observations add to the list of phenomena occurring during the critical phase early in training (see related work section 2), and prompts more investigations regarding this phase. In particular, it would be interesting to understand *why* the emergence of class-selectivity is necessary in this early phase of training, and whether this phenomenon connects to other critical-period phenomena, such as the emergence of lottery ticket subnetworks (Frankle & Carbin, 2018; Paul et al., 2022), and critical learning flexibility (Achille et al., 2018).

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

## A  APPENDIX

### A.1  MODEL TRAINING

We trained 10 instances of ResNet-50 on ImageNet using standard training procedure using PyTorch (Paszke et al., 2019). All instances were trained for 90 epochs with a batch size of 256, learning rate of 0.1, weight decay of 1e-4, and momentum of 0.9.

The class selectivity indices were calculated over the validation set of 50k images for every epoch from epoch 0 to epoch 90. All plots were generated using Seaborn Library (Waskom, 2021). Accuracies shown in all plots are top-1 ImageNet training/validation accuracy. Error shades in all plots represent the 95% confidence interval (CI) of the mean.

### A.2  CLASS-SELECTIVE ABLATIONS ON A FULLY-TRAINED NETWORK ARE LESS DAMAGING THAN RANDOM ABLATIONS

We find that in the early and intermediate modules (modules 4, 5, 6), an ablation of units ordered by their class-selectivity rank (i.e. most class-selective units ablated first) affects the network accuracy less than a control experiment where we ablate channels in a random order (Fig 8), consistent with the findings of Morcos et al. (2018). This result is paradoxical, as it seems to indicate that although class-selective channels emerge during training in early modules, these channels do not support the decision of the fully-trained network. In the latter module (module 7), we find that class-selective channels do support the decision of the network more than random channels (Fig 8), consistent with our intuition (i.e. the layers near the end of the network do need to be class-selective for good performance) and previous findings by Morcos et al. (2018).

### A.3  CLASS SELECTIVITY INDEX ACROSS BOTTLENECK LAYERS

Fig 9 shows the evolution of class selectivity throughout training for every bottleneck layer inside each module. An interesting observation over here is that for module 6 and 7, the bottleneck layers are increasingly class-selective with network depth i.e, the latter bottleneck layers are more selective than the earlier ones inside those modules. For module 5, they exhibit roughly the same amount of selectivity later in training. However, for module 4, this trend is reversed and the bottleneck layers are decreasing class-selective with network depth.

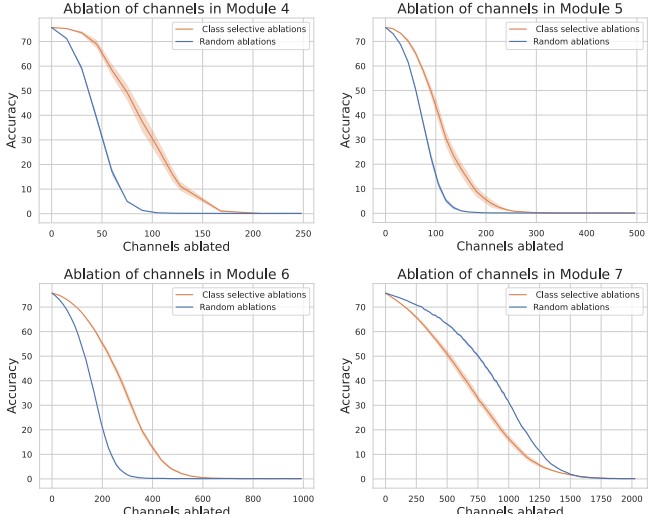

Figure 8: **Abation of channels in a fully-trained ResNet-50.** Surprisingly, ablating channels in a random order is more damaging to the network accuracy than ablating the most class-selective channels for modules 4, 5, 6. However, this trend is reversed for module 7. Error shades indicate the 95% CI of the mean, calculated over 10 networks trained from different random initial conditions.

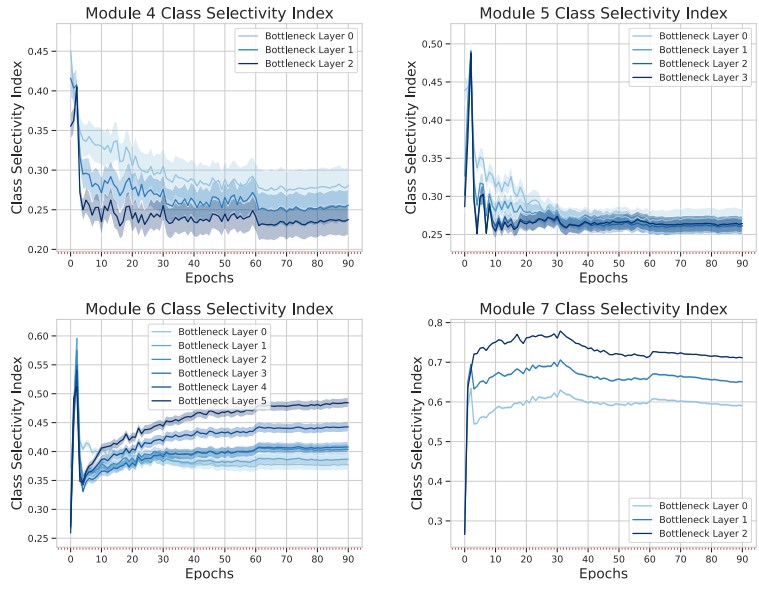

Figure 9: **Evolution of class selectivity throughout training across bottleneck layers.** Error shades indicate the 95% CI of the mean, calculated over the channels of each bottleneck layer.

## A.4 CENTERED KERNEL ALIGNMENT (CKA) ANALYSIS

To check the representational similarity between the modules at different stages of learning we used the Centered Kernel Alignment (CKA) (Kornblith et al., 2019; Nguyen et al., 2020) metric. To perform the CKA analysis we used the torch_cka library (Subramanian, 2021). The CKA score across all cases is high during the early epochs of training (Fig 10) which indicates that the network is jointly learning to be class-selective; with neurons from later layers relying on the class-selective features from early and intermediate layers to produce crude shortcut strategies.

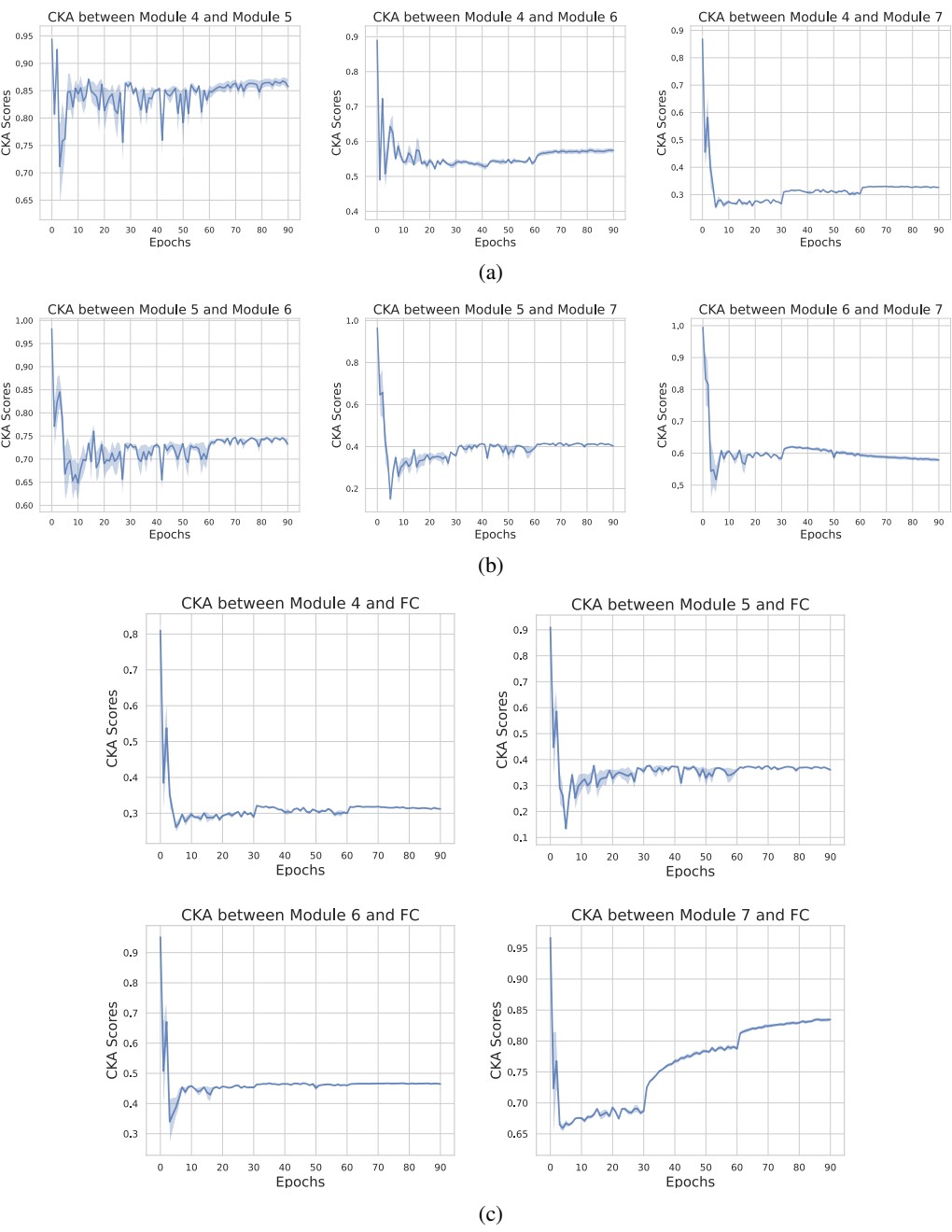

Figure 10: **CKA Analysis between different modules.** The representational similarity across all cases is high during the first few epochs after which it recedes to a low level for the rest of the training period. The high similarity in the initial epochs might explain the emergence of class-selective neurons as high similarity suggests the network is learning a crude, joint representation which causes the emergence of these class-selective neurons. Error shades indicate the 95% CI of the mean, calculated over the 6 different networks trained from different random initial conditions.

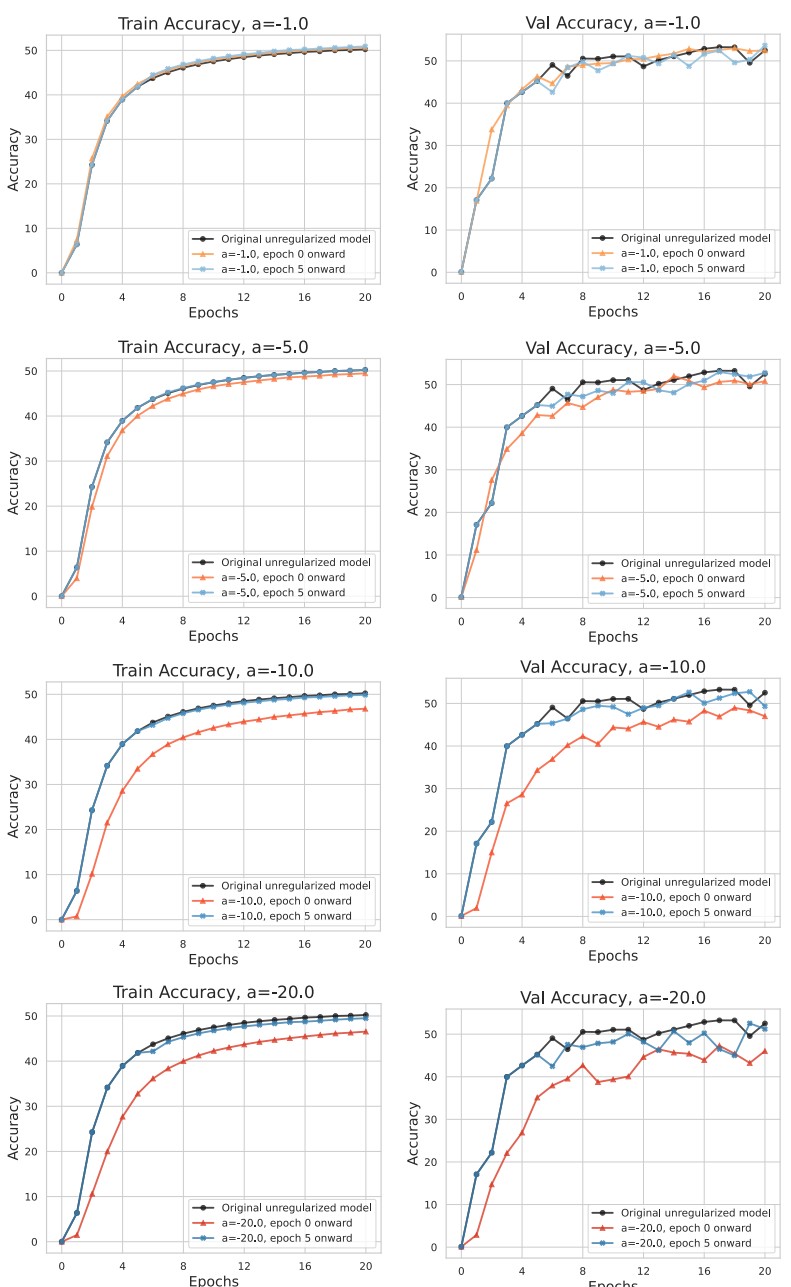

Figure 11: **Regularizing against selectivity for different values of $\alpha$ throughout training.**

Fig 10a shows that module 4 has a high representation similarity with module 5, 6, 7 in the first few epochs which then recedes to a low level for the rest of the training period. After the similarity level recedes, consistent our intuition, the similarity score decreases with increasing network depth i.e., after the first few epochs, module 4 is most similar to module 5, then to module 6, and then module 7. The remaining cases show a similar trend (Fig 10b, Fig 10c).

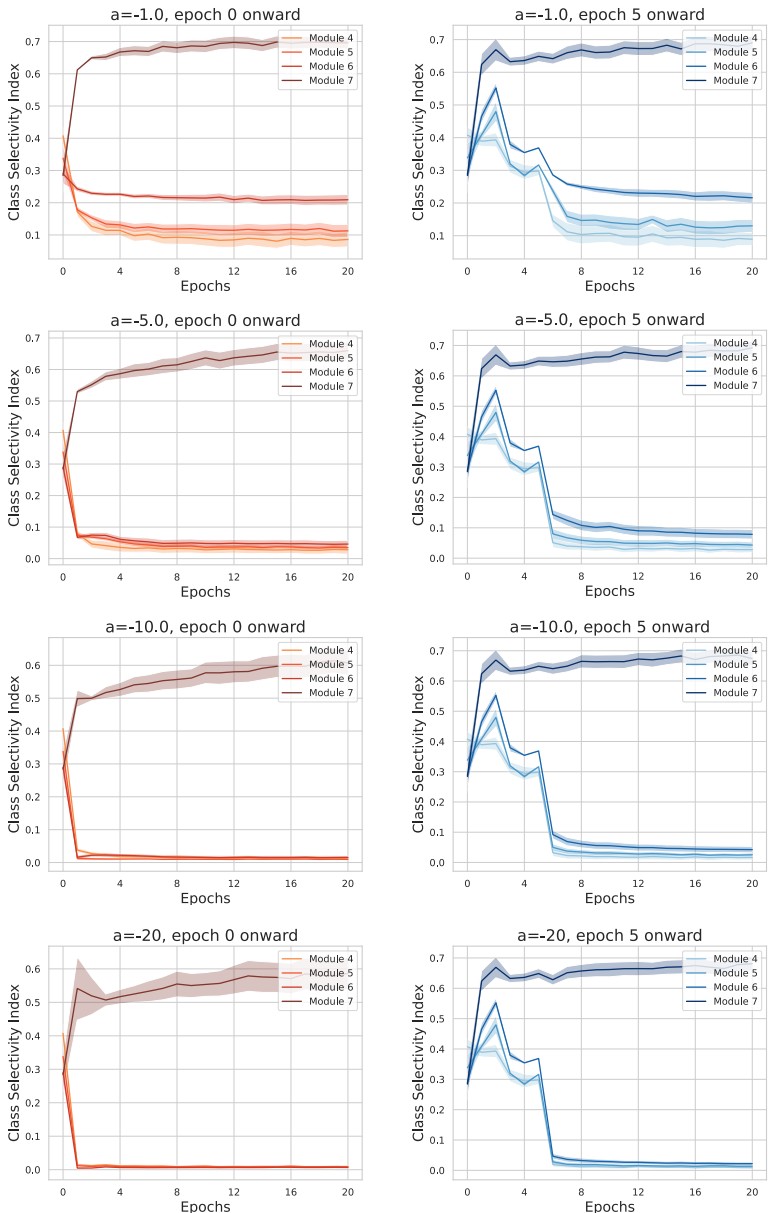

Figure 12: **Effect of regularization on selectivity indices across modules for different values of** $\alpha$ **throughout training.** Error shades indicate the 95% CI of the mean, calculated over the bottleneck layers of each module.

### A.5 Regularization Experiments

#### A.5.1 Regularizing against selectivity

We conduct a set of regularization experiments where we regularize against selectivity for different values of $\alpha$. We want to test two scenarios: Regularizing from epoch 0 onward VS regularizing from epoch 5 onward (epoch 5 is approximately after the spike in selectivity). We only regularize the early and intermediate modules (i.e. module 4, 5, 6) as we are interested in understanding the emergence of class selectivity in those modules. Also, regularizing only the early and intermediate modules allows us to use a higher values of $\alpha$ without squashing the network accuracy completely. The results for different values of $\alpha$ are shown in Fig 11. The following main observations can be made:

- For $\alpha$ = -1.0, there isn't much difference between the unregularized model and the models regularized from epoch 0 and epoch 5 onward.
- For $\alpha$ = -5.0, the model regularized from epoch 0 onward performed worse than the one regularized from epoch 5 onward.
- For $\alpha$ = -10.0, -20.0, the models regularized from epoch 0 onward performed *significantly* worse than the ones regularized from epoch 5 onward.

These results indicate that selectivity is important for learning in those early epochs and suppressing it by a significant amount impairs learning.

#### A.5.2 Regularizing for selectivity

If selectivity is important for learning in the early epochs, then does increasing selectivity further in early epochs improve learning? To test this, we conducted experiments in which we regularized for selectivity with different values of $\alpha$. First, we tried regularizing the early and intermediate modules with large values of $\alpha$ (+10, +20) for the first four epochs, after which the regularizer is turned off. The latter module (module 7) was not regularized for selectivity. We found that in both cases, the model performed much worse as compared to the original unregularized model (Fig 13). The large $\alpha$ causes the selectivity index to go close to 1.0. Also, as module 7 was not regularized, its selectivity index ends up being lower than the other modules (Fig 13).

Next, we re-did the experiment with smaller values of $\alpha$ (+1, +5) and we also decided to regularize module 7 for selectivity along with the early and intermediate modules. This time, we tried the experiment on two scenarios:

- Turning the regularizer on for first four epochs for all modules with $\alpha$ = +1, +5.
- Turning the regularizer on only for epoch 1 (as it is the most important (Section 4.5)) for all modules with $\alpha$ = +1, +5.

In all cases, the increase in selectivity was harmful to performance (Fig 14). So overall, the results show that increasing selectivity does not improve performance. Therefore, both an increase and a decrease in selectivity is harmful to learning.

#### A.5.3 Can the model eventually recover its performance from the effects of regularization?

The regularization experiments conducted in Section 4.4 and Appendix A.5.1 are done till epoch 20. So, this naturally leads to the following questions:

**Does the model eventually recover from the effects of regularization if it is trained for more epochs?** We trained the models regularized from epoch 0 onward and the one regularized from epoch 5 onward for 60 epochs while keeping the regularizer on with $\alpha$ = -20. The model regularized from epoch 0 onward is never able to recover the lost performance (Fig 15a).

**Does the model eventually recover from the effects of regularization if the regularization is turned off after the early epochs?** Next, we regularized a third model for only the first four

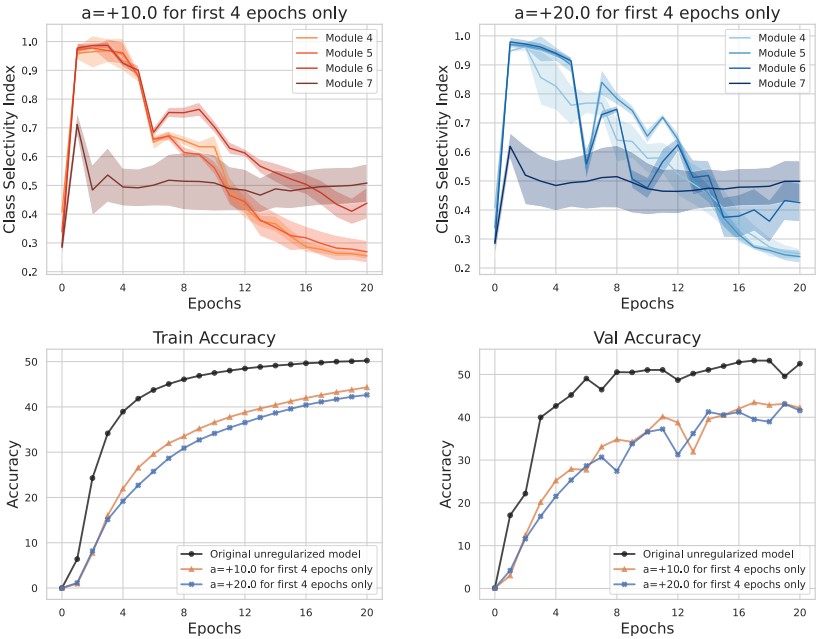

Figure 13: **Regularizing for selectivity with large values of $\alpha$**. Models regularized for selectivity with large $\alpha$ in the early epochs are harmful to performance. The regularization was performed on the early and intermediate modules (module 4, 5, 6) for first 4 epochs of training. Error shades indicate the 95% CI of the mean, calculated over the bottleneck layers of each module.

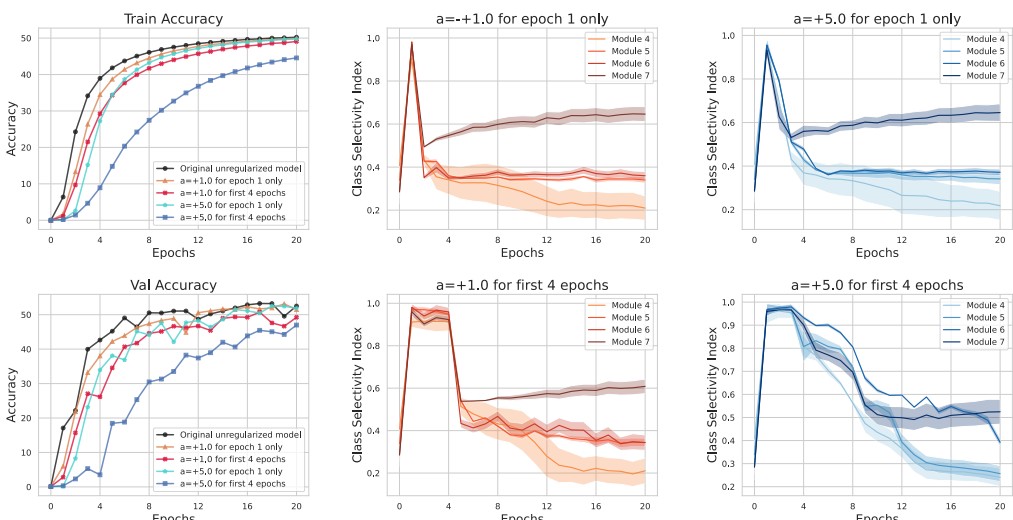

Figure 14: **Regularizing for selectivity with smaller values of $\alpha$.** Even models regularized with smaller values of $\alpha$ do not help in increasing the performance. The regularization was performed on all modules for two cases: Over first four epochs and only over the first epoch. Error shades indicate the 95% CI of the mean, calculated over the bottleneck layers of each module.

epochs with $\alpha$ = -20 after which the regularizer was turned off. This model was able to eventually recover the lost performance by epoch 20 (Fig 15b).

These two experiments show that class selectivity plays an important role towards model's performance.

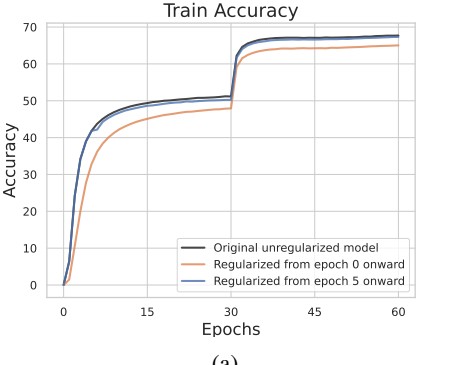 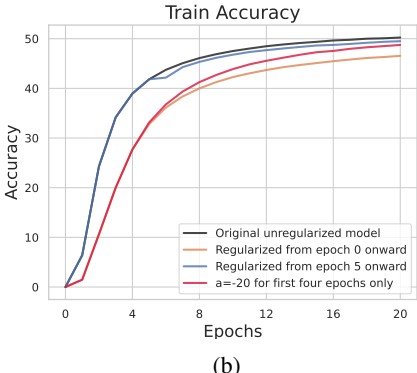

(a)               (b)

Figure 15: **Can the model recover from the effects of regularization? (a)** The models are trained longer with the regularizer kept on at $\alpha$ = -20, but the model trained from epoch 0 is never able to recover. **(b)** If the regularizer is only kept on for only first four epochs then the model manages to recover from the effects of regularization.

### A.5.4 EFFECT OF REGULARIZATION ON BALANCE OF CLASS REPRESENTATION

What happens to the class representation when we regularize against selectivity? To test this, we kept a count of each class predicted by the model on the validation set for each epoch, and then took the mean of top-5 class counts for each epoch. One would expect that if the model is regularized against selectivity then classes won't be over-represented i.e., the mean of top-5 class counts should be low for each epoch. However, surprisingly, the mean of top-5 class counts is actually higher in case of the model regularized from epoch 0 when compared to the unregularized model (Fig 16). So even if the individual neurons aren't class-selective, the model ends up over-representing certain classes in the early epochs. We think this might be because if neurons are not class-selective at all, then it might have difficulty learning in the early epochs and so the model as a whole ends up overfitting to a few classes.

Similarly, when the model is regularized in favor of selectivity, the classes again end up being over-represented. So this suggests that tampering with the selectivity in either direction perturbs the correct learning by over-representing classes.

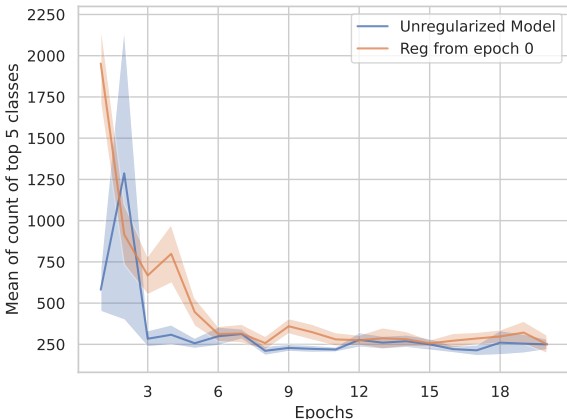

Figure 16: **Class representation over epochs.** The figure shows the mean of top-5 class counts, where the class counts are counted as the classes predicted by the model over the validation set in each epoch. The model regularized from epoch 0 over-represents classes in the early epochs. Error shades indicate the 95% CI of the mean, calculated over the top-5 mean count of two different networks trained from different random initial conditions.

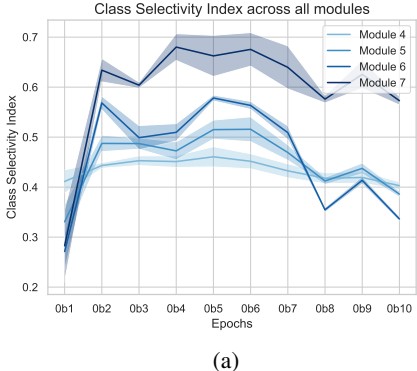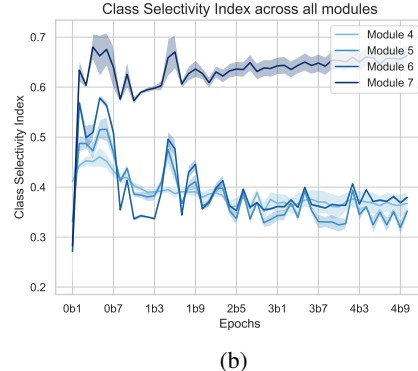

(a)                  (b)

Figure 17: **Evolution of class selectivity at a sub-epoch resolution.** The model is saved every 1000 batches of training within an epoch and the labels on the x-axis are of the form (e)b(i) where e represents the epoch number and i represents the batch number (in multiples of 1000). **(a)** The selectivity index rises for all modules after training the model on first 2000 batches. **(b)** Selectivity is most prominent during the first few thousand batches after which it starts to settle down.

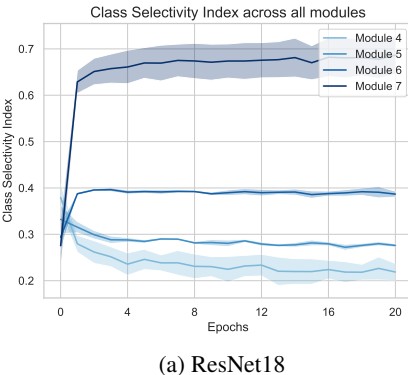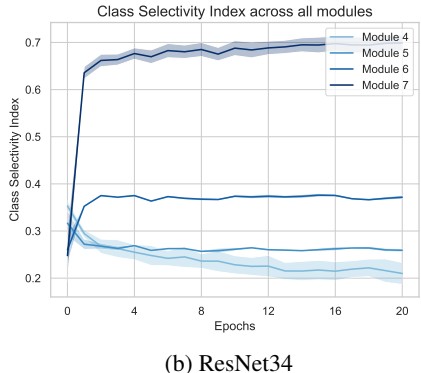

(a) ResNet18                 (b) ResNet34

Figure 18: **Evolution of class selectivity throughout training in all modules of ResNet18 and ResNet34**

## A.6    Analyzing class selectivity at a sub-epoch resolution

To understand how soon class selectivity rises during the first epoch of training, we analyzed selectivity at a sub-epoch resolution (Fig 17). The model was saved 10 times within each epoch, i.e, the model was saved after every 1000 batches of training with a batch size of 128, for the first few epochs. Fig 17a shows the evolution of selectivity during the first epoch of training. We can see that the selectivity rises after training the model on the first 2000 batches (Point 0b2 in Fig 17a). Fig 17b shows the evolution of selectivity at a sub-epoch resolution from epoch 0 to epoch 4. We can observe that the selectivity is most prominent during the first few thousand batches of training after which it starts to settle down.

## A.7    Class Selectivity in smaller variants of ResNets

In this section, we investigate whether smaller variants of ResNet (i.e. ResNet18/34) also exhibit similar class selectivity characteristics as a ResNet50. The ResNet18 and ResNet34 variants can also be divided into four modules like the ResNet50 (Fig 1). Both ResNet18 and ResNet34 have fewer layers and also fewer channels in each layer compared to a ResNet50.

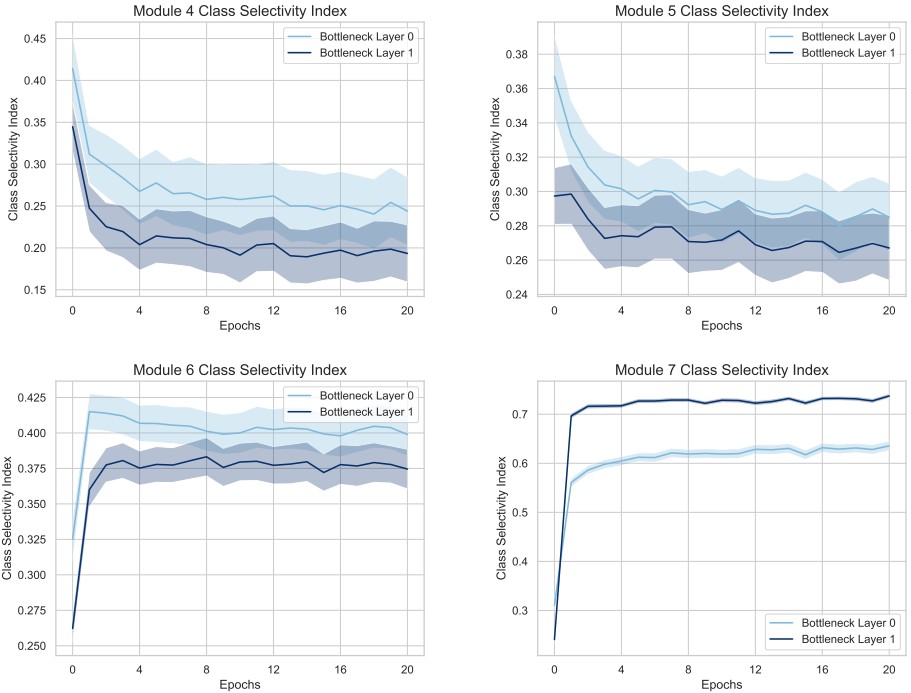

Figure 19: **Evolution of class selectivity throughout training across bottleneck layers for ResNet18**

### A.7.1 EVOLUTION OF CLASS SELECTIVITY IN ALL MODULES

In both ResNet18 and ResNet34, the rise in selectivity in early epochs can only be observed for module 6 and 7 (Fig 18). The selectivity values then stabilize across all modules just like what was observed in the case of ResNet50. Another interesting observation is that the stabilized class selectivity values for each module are approximately the same across all three ResNet architectures, i.e, for module 7 it is $\approx 0.7$, for module 6 it is $\approx 0.4$, for module 5 and 4 it is $\approx$ between 0.2-0.3.

The lower rise in selectivity could possibly be because of smaller depth (fewer layers) and also significantly fewer channels in each layer. For comparison, in a ResNet50 there are 256, 512, 1024, 2048 channels in each layer of module 4, 5, 6 and 7 respectively while in ResNet18/34 there are only 64, 128, 256, 512 channels in each layer of module 4, 5, 6 and 7 respectively.

### A.7.2 EVOLUTION OF CLASS SELECTIVITY IN BOTTLENECK LAYERS

The evolution of selectivity in bottleneck layers of ResNet18 and ResNet34 (Fig 19, Fig 20) show a similar trend to that of ResNet50 bottleneck layers (Fig 9). For ResNet34, modules 6 and 7 are increasingly class-selective with network depth and modules 4 and 5 are decreasingly class-selective with network depth (Fig 20). The same effect can be observed for ResNet18 except for module 6 which is decreasingly class-selective with network depth (Fig 19).

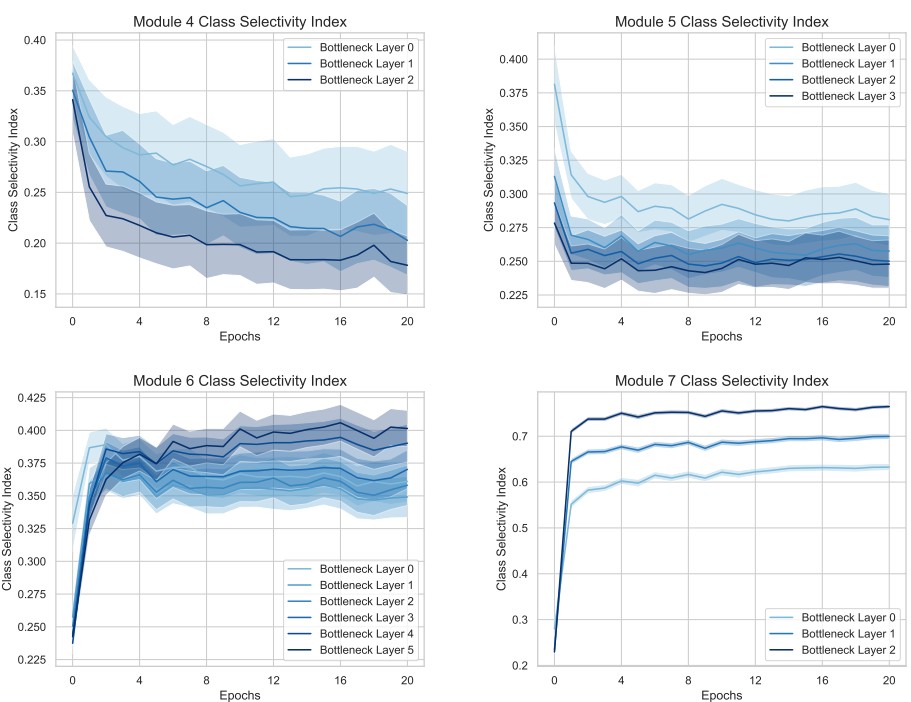

Figure 20: **Evolution of class selectivity throughout training across bottleneck layers for ResNet34**

