# OpenReview forum: "Importance of Class Selectivity in Early Epochs of Training"
_ICLR.cc/2023/Conference — Submitted to ICLR 2023_

### Official Review · Reviewer_1wf7 · 2022-10-20

**Confidence:** 3
**Correctness:** 2
**Technical Novelty And Significance:** 2
**Empirical Novelty And Significance:** 2
**Recommendation:** 5

**Clarity, Quality, Novelty And Reproducibility:**

The clarity should be improved. More experimental details should be introduced

The quality, novelty, and reproducibility are good for this paper.


**Strength And Weaknesses:**

Strength: The research topic is important. What happens in the early learning phase is still known little.

Weakness:

1. As the authors discussed, the main limitation of this paper only provides experiments on ResNet-50 and ImageNet. More experiments with more kinds of CNNs and datasets are necessary. Otherwise, many questions may occur, e.g., if using a larger CNN with more redundant parameters, does the phenomenon exist?
2. The authors only use one kind of learning rate to conduct experiments. Different optimizations should be adopted to check whether the phenomenon is from the optimization.
3. More explanations about experiments and metrics for Figure 3 and 4 are needed.
4. In Figure 6 (b) and 7, the authors only provide results of 20 epochs, without standard deviation. With more punishment from regularization, the learning speed should be slower than others at the beginning. After 20 epochs, will the experiment with regularized from epoch 0 catch up or be close to others?


**Summary Of The Paper:**

In this paper, the authors research the effects of class-selective neurons in the early learning phase. And the authors design different experiments for the claims, class-selective neurons are important for networks during the early training phase, the early and later layers are similar in the early training phase, and class-selective neurons in the early and intermediate layers are essential to the successful training of the network in the first few epochs.

**Summary Of The Review:**

Although the research topic is interesting, experiments are not sufficient for supporting claims. I do not recommend this work for the current version.

---

> ### Author Response · Authors · 2022-11-18
> **Response to Reviewer 1wf7**
>
> 1. **As the authors discussed, the main limitation of this paper only provides experiments on ResNet-50 and ImageNet. More experiments with more kinds of CNNs and datasets are necessary. Otherwise, many questions may occur, e.g., if using a larger CNN with more redundant parameters, does the phenomenon exist?** \
> => We are in the process of reproducing our main results (dynamics of the emergence of class-selective neurons in intermediate layers, effect of regularization against class-selectivity during early epochs of training) on a ResNet-18 and a ResNet-34. ViT is a good idea, but we did not have the time to run the experiment by the deadline. We will upload a new version of the paper with the extra-experiments by the Friday 18th deadline.
>
> 2. **The authors only use one kind of learning rate to conduct experiments. Different optimizations should be adopted to check whether the phenomenon is from the optimization.** \
> => Leavitt and Morcos ran extensive parameter sweeps across different models and parameters, and they did find reliable results across all these conditions. We believe that our results would also be robust to these different parameters, but we did not have the computational resources to perform an extensive parameter sweep in this study.
>
> 3. **More explanations about experiments and metrics for Figure 3 and 4 are needed.** \
> => We have now added details and more explanations on the class-selectivity metric used in Figure 3 and 4 and we hope that it addresses the clarity concerns of the reviewer.
>
> 4. **In Figure 6 (b) and 7, the authors only provide results of 20 epochs, without standard deviation. With more punishment from regularization, the learning speed should be slower than others at the beginning. After 20 epochs, will the experiment with regularized from epoch 0 catch up or be close to others?** \
> => We have investigated these questions in Appendix A.5.3. We found that if the regularizer is kept on then the model regularized from epoch 0 onward never manages to catch up to the model regularized from epoch 5 onward (Fig 15a). However, if the regularizer is turned off after the first 4 epochs, then the model manages to recover from the effects of the regularization in the latter epochs (Fig 15b). .

---

> > ### Comment · Reviewer_1wf7 · 2022-11-21
> > **Response to Authors**
> >
> > 1. The results of ResNet-18 and ResNet-34 are welcome. I suggest authors should modify abstract.
> >
> > 2. If the regularization is removed after the first few epochs (figure 15), the performance of the network can catch up in the later epochs, which is against the main claim that the emergence of the class-selective neurons during the first few epochs is essential to the successful training.

---

> > > ### Author Response · Authors · 2022-11-23
> > > **Response to Reviewer 1wf7**
> > >
> > > * **If the regularization is removed after the first few epochs (figure 15), the performance of the network can catch up in the later epochs, which is against the main claim that the emergence of the class-selective neurons during the first few epochs is essential to the successful training.** \
> > > => We thank the reviewer for noting an ambiguity in one of our claims and we will be updating this claim in the camera-ready version for clarity. The regularization experiments do show that class selectivity is important during the first epochs of training, as suppressing selectivity during the first epoch impacts learning, while suppressing selectivity later has no noticeable effect on learning. In that sense, our experiments reveal that the emergence of selectivity plays an important role in the early epochs, *in a standard network training scenario* (i.e. one without any explicit class-selective regularization). However, as the reviewer points out, if we turn-off the regularizer after the first few epochs and allow class-selective neurons to emerge then the network can eventually and slowly gain back the performance lost. In order to make our claims clearer, we will replace the ambiguous claims in the article with the following unambiguous statement: **"Class-selective neurons play a distinctively useful role during the first epoch of training, after which they can be suppressed without any noticeable effect on accuracy."**

---

> > > > ### Comment · Reviewer_1wf7 · 2022-11-23
> > > > **Response to Authors**
> > > >
> > > > ""Class-selective neurons play a distinctively useful role during the first epoch of training, after which they can be suppressed without any noticeable effect on accuracy"
> > > >
> > > > From my understanding, the regularization is to suppress the class-selective neurons. The experiment was conducted with regularization, then the regularization was removed after the first four epochs. So, the experiment shows that class-selective neurons can be suppressed at the beginning without any noticeable effect on accuracy. In other words, class-selective neurons in the early phase of training are not important for the successful training.

---

> > > > > ### Author Response · Authors · 2022-11-26
> > > > > **Response to Reviewer -  1wf7**
> > > > >
> > > > > The reviewer is correct to point out that class-selective neurons can be killed during the early epochs of training and that the network can eventually and slowly recover from that perturbation if class-selective neurons are allowed to emerge after that. However, in this scenario we are modifying the standard training procedure and changing the learning dynamics by suppressing and then unsuppressing class selectivity using the regularizer.
> > > > >
> > > > > We aren’t claiming that the network is incapable of gaining back the performance if class selectivity is allowed to emerge later on. The purpose of our experiments was to study why and how class selectivity emerges due to the learning dynamics of the standard training procedure. Our other experiments conducted using unregularized, standard training procedure have shown that selectivity rises in the early epochs and this is most likely due to the high linearity of the network and therefore, in any standard training procedure selectivity should rise in the early epochs due to the unique dynamics of those epochs. The purpose of our regularization experiments was to study whether these class-selective neurons which occur naturally in these early epochs are important for learning. The regularization experiments showed that indeed class-selective neurons are useful as demonstrated by the very different learning dynamics happening when they are regularized in the early epochs (here useful does not mean irreplaceable). Therefore, we believe that in this standard scenario, our claim about class-selective neurons being helpful to learning in the early epochs holds true. The reviewer is technically correct to point out that when we change this standard scenario by regularizing and then unregularizing, then the network can eventually gain back performance but that doesn’t contradict our claim in the standard training procedure (i.e. the scenario in which class-selective neurons naturally rise in the early epochs).
> > > > >
> > > > > However, we do agree that the way we worded our claim was ambiguous. Therefore, we propose this more technical claim to replace the previously proposed claim in the abstract, intro and corresponding result section in the final version of the paper: *"We find that suppressing class-selective neurons in early epochs of training drastically perturbs learning dynamics, whereas this suppression doesn’t have any noticeable effect on learning dynamics after the first few epochs of training.”*

---

### Official Review · Reviewer_TZoE · 2022-10-23

**Confidence:** 4
**Correctness:** 4
**Technical Novelty And Significance:** 3
**Empirical Novelty And Significance:** 3
**Recommendation:** 6

**Clarity, Quality, Novelty And Reproducibility:**

The paper is of very good clarity and quality. In particular, the paper clearly states the objective (ResNet-50s, class selectivity in the temporal dimension) to study, and the model component (bottleneck layers) for analysis. During analysis, the authors proposed concrete method and metric to quantify the class selectivity change and their explanations.

The analysis is also original, which not only considered selectivity across layers, but also considered change across training epochs. Using regularization starting at various epochs is also new.

**Strength And Weaknesses:**

The strength of the paper lies mainly in the thoroughness of the analysis. The author separates the research into two aspects: empirical observations (and the tentative explanation), and their role in model training.

There are several weakness of the paper:
1) The limitation of the model backbone. They only consider ResNet-50s, but not other ResNet (such as ResNet-152), or other architecture (ViT). It will be much more interesting to see how different model contributes to the effect of class selectivity.
2) For image classification, the paper restricts to ImageNet only. ImageNet is a relatively small sized data and there is a good notion of "epochs", but in real practice once can encounter data that takes a long time to train to one epoch. How does class selectivity behave for those large dataset within one epoch?
3) How about other tasks beyond vision? Such as NLP, speech, etc?
4) Currently, the paper is mostly about empirical observations; have the author thought about how to utilize the observation on class selectivity to improve the model training? The improvement can be in terms of accuracy, robustness, etc.

**Summary Of The Paper:**

The paper studies in a systematic way the effects of "class selectivity" in early epochs of training.
1) The main backbone architecture is ResNet-50 on ImageNet and their early / intermediate / latter modules. Units (or neurons) are defined as individual channels of bottleneck layers in each module.
2) For each unit, the author used "class selectivity index" defined in previous literature to study how "special" the unit reacts to each ImageNet class.
3) The first part of the main paper is an observation that class selectivity appears strongly in all modules in early epochs. After that, the early and intermediate layers saw decrease in the selectivity while latter layers there does not seem to be a decrease. The author also gives a tentative explanation that the network is approximately linear so all modules behave similarly.
4) The second part of the main paper discusses the importance of class selectivity, by applying regularization to either suppress or encourage selectivity, from the first epoch or from later epoch. The author concludes that the selectivity in early epochs is very important for quality of training.

**Summary Of The Review:**

I recommend the paper with marginally acceptance for two reasons: 1) The analysis on temporal change is new and comprehensive. 2) The study itself is, however, too narrow. ResNet-50 is no longer the most up-to-date architecture, and ImageNet is a well-curated dataset where class distribution is relatively balanced. So there is still questions regarding how general this phenomenon could be. On the other hand, the authors didn't proceed with how to use the observation to motivate a better training procedure.

---

> ### Author Response · Authors · 2022-11-18
> **Response to Reviewer TZoE**
>
> 1. **The limitation of the model backbone. They only consider ResNet-50s, but not other ResNet (such as ResNet-152), or other architecture (ViT). It will be much more interesting to see how different model contributes to the effect of class selectivity.** \
> => We are in the process of reproducing our main results (dynamics of the emergence of class-selective neurons in intermediate layers, effect of regularization against class-selectivity in early epochs of training) on a ResNet-18 and a ResNet-34. ViT is a good idea, but we did not have the time to run the experiment by the deadline. We will upload the paper with the extra-experiments by the Friday 18th deadline.
>
> 2. **For image classification, the paper restricts to ImageNet only. ImageNet is a relatively small sized data and there is a good notion of "epochs", but in real practice once can encounter data that takes a long time to train to one epoch. How does class selectivity behave for those large dataset within one epoch?** \
> => We don’t think epochs are the critical timescale here. To demonstrate this, we have added a new experiment where we show that the spike in class-selectivity occurs during the first 2000 batches of training (compared to 9000 batches total in one epoch). See Fig 17 in Appendix A.6. We would like to investigate larger datasets but we are unable to investigate it given the time and computational resources we have during this rebuttal period.
>
> 3. **How about other tasks beyond vision? Such as NLP, speech, etc?** \
> => Interesting suggestion but beyond the scope of what we can do.
>
> 4. **Currently, the paper is mostly about empirical observations; have the author thought about how to utilize the observation on class selectivity to improve the model training? The improvement can be in terms of accuracy, robustness, etc.** \
> => We have tried to increase, decrease class selectivity in early epochs but none of our attempts improved performance (See Appendix A.5.1, A.5.2). In future work, we could also test for robustness to OOD datasets, regularizing from epoch 1 onward. Prior work has indeed shown that regularizing against class selectivity can improve robustness to OOD samples (Leavitt & Morcos, 2021: Linking average- and worst-case perturbation robustness via class selectivity and dimensionality), and the knowledge that class selectivity is only critical during the first epoch of training could be leveraged to design a new regularzing curriculum.

---

> > ### Comment · Reviewer_TZoE · 2022-11-21
> > **Thanks for the clarification**
> >
> > Thanks the author for the clarifications. The rating stays the same.

---

> > > ### Author Response · Authors · 2022-11-23
> > > **Thank you**
> > >
> > > Thank you for taking the time to review our paper and for the insightful suggestions!

---

### Official Review · Reviewer_wgvT · 2022-10-25

**Confidence:** 3
**Correctness:** 3
**Technical Novelty And Significance:** 1
**Empirical Novelty And Significance:** 2
**Recommendation:** 6

**Clarity, Quality, Novelty And Reproducibility:**

This work is strong in terms of clarity, quality, and reproducibility. However, there is little novelty in terms of technical innovations.

**Details Of Ethics Concerns:**

There are no ethical concerns.

**Strength And Weaknesses:**

Strengths:

- Writing: The paper is clearly written and easy to follow.
- Insight: The paper does provide some new insight regarding the usefulness of class selectivity in early epochs of training.

Weaknesses:

- Novelty: There is little technical novelty in this paper.
- Significance: While the paper does provide some insight regarding class selectivity, I fail to how this insight may benefit us when we train neural networks.

**Summary Of The Paper:**

In this paper, the authors conducted a study on the importance of class-selective neurons in early and intermediate neurons during the training process. For their study, the authors trained a ResNet-50 on ImageNet, and recorded the class selectivity index of neurons in modules 4 through 7 of the ResNet-50 during training. They found that class selectivity of early and intermediate neurons generally increases during the first few epochs of training, and rapidly decreases after the first few epochs. The authors also regularized for/against class selectivity of early and intermediate neurons. They found that regularizing against class selectivity during early epochs of training significantly hampers network training.

**Summary Of The Review:**

Based on the strengths and weaknesses discussed above, I believe that this paper has some value, but is lacking in terms of technical novelty and significance.

************************************
After reading the authors' response, I decided to increase my rating to 6 ("marginally above the acceptance threshold"), because the authors did explain how their study could be potentially used to improve neural network training. I do encourage authors to incorporate their response into the main paper.

---

> ### Author Response · Authors · 2022-11-18
> **Response to Reviewer wgvT**
>
> 1. **Novelty: There is little technical novelty in this paper.** \
> => We agree that this work does not introduce groundbreaking methods or technical approaches. We believe that the value of our work comes from answering questions related to *why* neural networks learn class selectivity, and whether it is necessary for learning high-quality solutions. Class selectivity is widely used in interpretability research (Zhou et al., Object Detectors Emerge in Deep Scene CNNs, 2015; Olah et al., Feature Visualization, 2017; Radford et al., Learning to Generate Reviews and Discovering Sentiment, 2017; Olah et al., The Building Blocks of Interpretability, 2018; Hooker et al., A Benchmark for Interpretability Methods in Deep Neural Networks, 2019; Na et al., Discovery of Natural Language Concepts in Individual Units of CNNs, 2019), and has been extensively studied with regards to generalization and network performance (Morcos et al., On the importance of single directions for generalization, 2018; Amjad et al., Understanding Individual Neuron Importance Using Information Theory, 2018; Zhou et al., Revisiting the Importance of Individual Units in CNNs via Ablation, 2018; Kanda et al., Deleting object selective units in a fully-connected layer of deep convolutional networks improves classification performance, 2020; Dalvi et al., What Is One Grain of Sand in the Desert? 2019; Donnelly and Roegiest, On Interpretability and Feature Representations: An Analysis of the Sentiment Neuron, 2019; Meyes et al., Ablation Studies in Artificial Neural Networks 2019). Existing evidence regarding the necessity of class selectivity for high-quality solutions is mixed—some in favor, some not, and our work reveals nuances that resolve some of these contradictions in the literature (for example by showing that class selectivity is necessary during the early phase of training).
>
> 2. **Significance: While the paper does provide some insight regarding class selectivity, I fail to how this insight may benefit us when we train neural networks.** \
> => Although we have not demonstrated an immediate benefit of our findings in this paper, there are some novel empirical findings which may benefit us as if we understand the critical period of early training better, we might understand how to deal with training better. Prior work shows that regularizing against class selectivity improves robustness to OOD tasks (Leavitt & Morcos, 2021: Linking average- and worst-case perturbation robustness via class selectivity and dimensionality). One could potentially use our findings to design a curriculum of regularization which would not prevent training during the early critical period where class selectivity is useful, but would improve OOD robustness during the rest of training.

---

> > ### Comment · Reviewer_wgvT · 2022-11-21
> > **Thank you for your clarification**
> >
> > Thank you for your clarification! After reading the response, I have increased my rating to 6 ("marginally above the acceptance threshold").

---

> > > ### Author Response · Authors · 2022-11-23
> > > **Thank you**
> > >
> > > Thank you for the valuable feedback and for updating the score!

---

### Official Review · Reviewer_vP2D · 2022-11-02

**Confidence:** 4
**Clarity, Quality, Novelty And Reproducibility:** Quality, clarity and originality of t…
**Correctness:** 3
**Technical Novelty And Significance:** 2
**Empirical Novelty And Significance:** 3
**Recommendation:** 6

**Strength And Weaknesses:**

# Strengths
- It's nice to see the specialization of neurons in earlier layers being investigated. Though initial results doesn't lead to better algorithms, it could help us understand training and eventually help us improve optimization of NNs.

# Weakness
- It would help to check whether findings transfer to a different architecture / dataset. And obviously one doesn't need to repeat all experiments. A single plot combining different metrics could suffice (see suggestions). Maybe a ViT?

[During rebuttal] Authors said no-time (which is understandable)

- I couldn't see a motivation for the importance of studying neuron selectivity. Do we want/expect neuron selectivity to be 0? Why? I think this is a great exploratory study but I don't it is well motivated why this is important for the community.

[During rebuttal], authors pointed out previous work who uses this metric in different settings (most of which already cited in paper). I don't think this is enough for motivating a research work. Why do they use it? Why not another metric (we can come up with many metric like this)? Why we should look at this phenomenon and not others? Addressing these questions would improve the work (instead of saying this metric is being used in previous work)

- I'm not sure adding an additive regularizer for reducing neuron selectivity makes the results/relation causal. This could be solely about optimization. One way to get a very low neuron selectivity is to push all weights to zero and thus all zero activations and this could prevent/effect later learning. However, it doesn't mean that reducing neuron sensitivity always lead to worse generalization. You also probably want to have different coefficients for different layers as we want/expect class sensitivity in later layers.

[During rebuttal] Thanks for sharing the difference between previous work and regularization used. It might worth mentioning this in the paper (if it is not there already) I still believe using `causal role` and `causal experiment` can be miss-interpreted.

# Suggestions
- I think it would help to story to have a plot where first 10 epoch of the training is highlighted with more frequent data points. It would be also nice to look at the Class sensitivity and CKA score AOC in the same plot to see the correlation more clearly.
- I think it would be nice to track the linear-probe performance of different layers over the course of the training. It would be nice to see whether that correlates with the Class-sensitivity metric. I think previous work on early-exit of NNs had some experiments on this, but only looked at things at the end of the training.

# Minor
- [First sentence of the paper] uses `neuron selectivity`. It would be nice to explain what neuron selectivity is or use a high level description ("understanding the role of neurons").
- [Section 3.2] It would be nice to define the index using a more precise notation. For example: would the stem convolutional layer with kernel shape (3x7x7x64) have 64 neurons in your notation? Are the activations assumed to be non-negative (after relu)?
- Figure 2 appears after 3 and 4.

# After Rebuttal
- I thank the authors for their response. I keep my score as it is, as my concerns for "why not higher?" stays the same. I think this is an interesting work and I support acceptance. Regardless I encourage authors to improve their work in camera ready or for next submission using the feedback given.


**Summary Of The Paper:**

This paper studies the "class sensitivity" of a ResNet-50 backbone during its training on Imagenet. "Class sensitivity" (defined in previous work) characterizes whether a given neuron fires disproportionally for a particular class. They also study the effect of ablating (pruning) class sensitive neurons on final accuracy an observe they are more prunable than a randomly selected neuron. Main results show that (1) Class sensitivity increases during the first few epochs of training and reduces (2) During this initial phase pruning highly class sensitive neurons incur higher damage compare to later epochs (still less than random). (3) higher similarity between certain layers during the early phase.

**Summary Of The Review:**

Overall this work is a potentially a useful study for explaining the evolution of neural networks during the early phase of training. The observation that early layers learn to class-sensitive neurons together with the high similarity between layers suggest a kind of shortcut learning which to my knowledge a novel observation and would be beneficial to the community. My concerns lie around the (a) experimental evaluation (limited to a single setting) (b) motivation (c) 4th claim/contribution. If addressed, would be willing to change my evaluation.

---

> ### Author Response · Authors · 2022-11-18
> **Response to Reviewer vP2D - Part 1**
>
> 1. **It would help to check whether findings transfer to a different architecture / dataset. And obviously one doesn't need to repeat all experiments. A single plot combining different metrics could suffice (see suggestions). Maybe a ViT?** \
> => We are in the process of reproducing our main results (dynamics of the emergence of class-selective neurons in intermediate layers, effect of regularization against class-selectivity in early epochs of training) on a ResNet-18 and a ResNet-34. ViT is a good idea, but we did not have the time to run the experiment by the deadline. We will upload the paper with the extra experiments by the Friday 18th deadline.
>
> 2. **I couldn't see a motivation for the importance of studying neuron selectivity. Do we want/expect neuron selectivity to be 0? Why? I think this is a great exploratory study but I don't it is well motivated why this is important for the community.** \
> => We believe that the value of our work comes from answering questions related to \emph{why} neural networks learn class selectivity, and whether it is necessary for learning high-quality solutions. There is a significant existing body of research on selectivity: Class selectivity is widely used in interpretability research (Zhou et al., Object Detectors Emerge in Deep Scene CNNs, 2015; Olah et al., Feature Visualization, 2017; Radford et al., Learning to Generate Reviews and Discovering Sentiment, 2017; Olah et al., The Building Blocks of Interpretability, 2018; Hooker et al., A Benchmark for Interpretability Methods in Deep Neural Networks, 2019; Na et al., Discovery of Natural Language Concepts in Individual Units of CNNs, 2019), and class-selectivity has been extensively studied with regards to generalization and network performance (Morcos et al., On the importance of single directions for generalization, 2018; Amjad et al., Understanding Individual Neuron Importance Using Information Theory, 2018; Zhou et al., Revisiting the Importance of Individual Units in CNNs via Ablation, 2018; Kanda et al., Deleting object selective units in a fully-connected layer of deep convolutional networks improves classification performance, 2020; Dalvi et al., What Is One Grain of Sand in the Desert? 2019; Donnelly and Roegiest, On Interpretability and Feature Representations: An Analysis of the Sentiment Neuron, 2019; Meyes et al., Ablation Studies in Artificial Neural Networks 2019). Existing evidence regarding the necessity of class selectivity for high-quality solutions is mixed—some in favor, some not, and our work reveals nuances that resolve some of these contradictions in the literature, in particular by showing that class selectivity is necessary during the early phase of training.
>
> 3. **I'm not sure adding an additive regularizer for reducing neuron selectivity makes the results/relation causal. This could be solely about optimization. One way to get a very low neuron selectivity is to push all weights to zero and thus all zero activations and this could prevent/effect later learning. However, it doesn't mean that reducing neuron sensitivity always lead to worse generalization. You also probably want to have different coefficients for different layers as we want/expect class sensitivity in later layers.**  \
> =>  Yes, true, but what we find is that this perturbation affects performance in the first epochs but not after, which is quite interesting. It also matches with the timing in the spike in selectivity that we observe. We make the claim about causality because most previous studies have examined \emph{correlations} between network behavior and single unit selectivity (e.g. Bau et al., Network Dissection, 2017; Olah et al., The Building Blocks of Interpretability, 2018; Radford et al., Learning to Generate Reviews and Discovering Sentiment, 2017), for example by training a network, then looking at how selectivity correlates with some other measure of interest. Prior to Leavitt and Morcos’ (2020a) introduction of their selectivity regularizer, researchers had not directly, or causally, manipulated class selectivity \emph{during training}.
> Regarding your concern about a degenerate solution to reducing selectivity (“activation zeroing”), Leavitt and Morcos (2020a) show in Appendix Section A.11 of their paper that this does not seem to happen.
>
> 4. **I think it would help to story to have a plot where first 10 epoch of the training is highlighted with more frequent data points. It would be also nice to look at the Class sensitivity and CKA score AOC in the same plot to see the correlation more clearly.**  \
> => Thank you for this neat suggestion. We have now performed an analysis of class-selectivity in intermediate layers at a sub-epoch resolution during the first few epochs of training. We find that the sharp rise in class-selectivity occurs in all intermediate layers during the first 2000 batches of training (128 images per batch). We have now added this new analysis in the paper (see Fig 17 in the Appendix A.6).

---

> > ### Author Response · Authors · 2022-11-18
> > **Response to Reviewer vP2D - Part 2**
> >
> > 5. **I think it would be nice to track the linear-probe performance of different layers over the course of the training. It would be nice to see whether that correlates with the Class-sensitivity metric. I think previous work on early-exit of NNs had some experiments on this, but only looked at things at the end of the training.** \
> > => It would be a nice experiment to run but we didn’t have time. What we can say is that there is prior work that addresses a similar question. Wang et al., (Learning Robust Global Representations by Penalizing Local Predictive Power, 2019) introduce a training objective for image classification in CNNs that penalizes the predictive power of early layer(s) (in addition to the standard cross-entropy on the model outputs). In that case, where they regularize only one specific early layer of the network, they find that this results in an improvement on image classification performance.
> >
> > 6. **[First sentence of the paper] uses neuron selectivity. It would be nice to explain what neuron selectivity is or use a high level description ("understanding the role of neurons").**  \
> > => Thank you for the suggestion.
> >
> > 7. **[Section 3.2] It would be nice to define the index using a more precise notation. For example: would the stem convolutional layer with kernel shape (3x7x7x64) have 64 neurons in your notation? Are the activations assumed to be non-negative (after relu)?**  \
> > => You are correct that in your example, we would be measuring class-selectivity on 64 channels. We modified the text of section 3.2 to clarify these two points.
> >
> > 8. **Figure 2 appears after 3 and 4.** \
> > => Thank you, we corrected this.

---

### Decision · Program_Chairs · 2023-01-20

**Decision:**

Reject

**Justification For Why Not Higher Score:**

Please see meta review

**Justification For Why Not Lower Score:**

Reject

**Metareview: Summary, Strengths And Weaknesses:**

The paper investigates the importance of class selectivity during training.


Strength:
- Writing: The paper is clearly written and easy to follow.
- mainly in the thoroughness of the analysis for case of ResNet50 and imageNet dataset


Weakness:
- Limited novelty
- Limitation of model backbone and dataset, it needs to be checked whether findings transfer to a different architecture / dataset.
- The considered problem is not well motivated and reduces the significance of the work.

I encourage the authors to consider the received feedback for improving the paper and its future iteration.

**Summary Of Ac-Reviewer Meeting:**

We discussed the strengths and weaknesses of the paper and whether anyone would champion the paper for acceptance. None championed the work. Also, It became obvious that the limitation of architecture and dataset is the one that reviewers cannot ignore and therefore letting the authors add additional experiments in ample time and submitting to the next venue would make the most sense.